# Search for Osme Bonds with π Systems as Electron Donors

**DOI:** 10.3390/molecules29010079

**Published:** 2023-12-22

**Authors:** Xin Wang, Qingzhong Li, Steve Scheiner

**Affiliations:** 1The Laboratory of Theoretical and Computational Chemistry, School of Chemistry and Chemical Engineering, Yantai University, Yantai 264005, China; wangxin19980705@s.ytu.edu.cn; 2Department of Chemistry and Biochemistry, Utah State University, Logan, UT 84322-0300, USA

**Keywords:** σ-hole, NBO, AIM, SAPT, back donation

## Abstract

The Osme bond is defined as pairing a Group 8 metal atom as an electron acceptor in a noncovalent interaction with a nucleophile. DFT calculations with the ωB97XD functional consider MO_4_ (M = Ru, Os) as the Lewis acid, paired with a series of π electron donors C_2_H_2_, C_2_H_4_, C_6_H_6_, C_4_H_5_N, C_4_H_4_O, and C_4_H_4_S. The calculations establish interaction energies in the range between 9.5 and 26.4 kJ/mol. Os engages in stronger interactions than does Ru, and those involving more extensive π-systems within the aromatic rings form stronger bonds than do the smaller ethylene and acetylene. Extensive analysis questions the existence of a true Osme bond, as the bonding chiefly involves interactions with the three O atoms of MO_4_ that lie closest to the π-system, via π(C-C)→σ*(M-O) transfers. These interactions are supplemented by back donation from M-O bonds to the π*(CC) antibonding orbitals of the π-systems. Dispersion makes a large contribution to these interactions, higher than electrostatics and much greater than induction.

## 1. Introduction

Although noncovalent interactions are much weaker than covalent bonds, they occupy a vital position in chemistry, physics, biology, and other disciplines, including applications such as the construction of supramolecular structures [1,2], molecular recognition [3], the regulation of biomolecular structures and functions [4], availability in pharmacology for drug molecule adsorption [5], and molecular docking [6]. Ongoing research continues to identify new types of noncovalent interactions, the current list of which includes hydrogen bonds [7,8,9], triel bonds [10,11], tetrel bonds [12,13], phosphorus bonds [14,15], chalcogen bonds [16,17], halogen bonds [18,19,20], and so on. These bonds are of the general electron donor–acceptor sort, which involve atoms of Groups 1, 3, 4, 5, and 7 as acceptors. Electron-donating Lewis bases vary from conventional bases such as molecules with lone-pair electrons and anions to less conventional ones including π-systems, metal hydrides, radicals, and carbenes [21,22,23,24,25,26,27,28]. The σ-hole concept, originally developed by Politzer and coworkers to aid in the understanding of the halogen bond [29,30], has expanded since that juncture and serves as a central issue in many of these interactions. This idea was also extended to the closely related π-hole that typically lies directly above certain planar systems [31].

There are other transition metals that have been shown to participate in bonds of this sort. Karimi found a weak interaction between transition metals Au and Ge through theoretical and experimental studies [32]. After analyzing the X-ray structures of molybdenopterin and tungtrexin and conducting theoretical calculations, Bauzá et al. found that there was an attractive interaction between Mo or W atoms and Lewis bases, which they dubbed the Wolfium bond (WfB) [33]. In addition, other groups identified the Matere bond (MaB) as the interaction between the seventh subgroup elements—Mn, Tc, or Re—and Lewis bases [34].

A recent addition to this family of noncovalent interactions has been christened the Osme bond, which involves the Group 8 atoms Fe, Ru, Os, and Hs as primary electron acceptors. Unlike the majority of commonly observed noncovalent bonds, which are organized around a main group atom, a transition metal serves this function in so-called Osme bonds. Since the penultimate electronic layer of the transition metal is unsaturated, there are numerous valence states available to this atom. Recently, Daolio et al. discovered that the covalent O-M bond in the tetraoxide of the group 8 elements Fe, Ru, and Os induces a σ-hole along the bond extension, facilitating an attractive interaction with pyridine or pyridine *N*-oxide derivatives through theoretical and experimental methods [35]. OsO_4_ and RuO_4_ are often involved in special reactions in organic chemistry and used as reactants in some oxidation reactions. For example, OsO_4_ was used as an electrophilic reagent to react with ethylene to produce osme oxide heterocycles [36]. Due to its relatively high toxicity and expense limitations, Nicolaou et al. used *N*-methylmorpholin-*N*-oxide (NMO) and OsO_4_ as co-oxidants to form a catalytic cycling system to react with olefins, which has advantages of simplicity and economy [37]. To this point, there are only a few reports concerning the Osme bond, leaving a number of important issues unexplored.

π-electron systems play an integral role in many biological phenomena and have been the focus of much past research. Perhaps the most common of these phenomena is π-π stacking [38,39,40] which is associated with several important processes, such as protein folding, molecular recognition, and supramolecular self-assembly. In particular, some of the smaller electron-rich π structures, such as C_2_H_2_ and C_2_H_4_, have been frequently studied [41,42]. Zheng et al. studied C_2_H_4_/C_2_H_2_/C_6_H_6_/(CH_2_)_3_···MX (M = Cu, Ag, Au; X = F, Cl, Br, I) with wB97XD functional, determining the contribution of π···M bonds to the stability of two-body complexes [43]. Of particular interest, a search of the Cambridge Structure Database (CSD) identified several X-ray structures that appeared to contain what looked like an Osme bond [44,45], at least from a geometric point of view. That is, π-electron systems such as C_2_H_2_, C_2_H_4_, and C_6_H_6_ were observed in proximity to an Os atom and in a location that might present a σ-hole on the Os center.

The current work is designed to investigate this issue further and to explore the possibility of Os···π interactions that might fall into the category of a Osme bond. To this end, the tetrahedral MO_4_ unit, with M = Ru and Os, was considered a Lewis acid as the strong electronegativity of O ought to provide a healthy σ-hole on M. A range of different π-systems, which include the small C_2_H_2_ and C_2_H_4_, were compared to the more extended aromatic rings of C_4_H_5_N, C_4_H_4_O, and C_4_H_4_S. As described below, the nature of the binding between these two entities was examined in detail using quantum chemical calculations from a number of different perspectives, so as to ascertain the precise nature of their noncovalent interactions and to assess the contribution that might be made by a M···π Osme bond.

## 2. Results

The molecular electrostatic potential (MEP) of each monomer is exhibited in Figure 1 in the form of a color-coded mapping on the 0.001 a.u. isodensity surface. The two MO_4_ units are most negative near their O atoms, with a positive red region lying directly opposite each O, in what might be called a σ-hole. This hole is somewhat deeper over the larger Os atom, 0.062 a.u., vs. 0.055 a.u. for its lighter Ru congener. The six bases each have a negative blue region above the molecular plane, with a minimum above the center of roughly −0.025 a.u., although this value is slightly more negative for C_4_H_5_N and C_4_H_4_O. The nature of these potentials would suggest that the positive σ-hole of the MO_4_ ought to line up nicely if placed directly above the negative π-system of each base.

And such an alignment is indeed favored in heterodimers, as illustrated by the AIM diagrams in Figure 2. The energetics of each complex is reported in Table 1, along with the distance between the M atom, Ru or Os, and the center of each base unit. The interaction and binding energies are quite close to one another, indicating very low internal deformation of each monomer unit upon forming the complex. These small deformation energies are consistent with only very minor deviations in the internal MO_4_ angle from a perfect tetrahedron, as well as minimal bond length changes, described in great detail below. These dimerization energies are somewhat larger for Os than for Ru by 3.0–4.0 kJ/mol. C_2_H_2_ and C_2_H_4_ engage in the most weakly bound complex, with stronger bonds arising for the aromatic rings, most particularly C_4_H_5_N. Altogether, these interaction energies span a range between 9.5 and 26.4 kJ/mol.

The intermolecular distances in Table 1 refer to that between M and the center of each base, whether a C≡C or C=C bond in the first two, or the center of the ring in the aromatics. These distances lie between 3.55 and 3.78 Å. They are generally shorter for Ru than for Os, in keeping with the smaller size of the former, but not in all cases, such as C_2_H_2_ and C_2_H_4_ which are rather weakly bound for Ru.

The bond paths in the AIM diagrams of Figure 2 connect each of the three O atoms of the MO_4_ unit that are closest to the base to various atoms of the latter. What is notably absent is an intermolecular bond path involving the M atom, which would argue against the presence of a M···π Osme bond. This issue, as well as the nature of the intermolecular bonding, is explored via the data in Table 2. The first column displays the density of the bond critical point of each of the paths illustrated in Figure 2. Although the Os interaction energies are universally larger than their Ru analogues, this distinction is not reflected in ρ_BCP_, which is actually reversed in the aromatic cases.

The next four columns of Table 2 contain the E^(2)^ perturbation energies associated with the charge transfer between particular pairs of NBO orbitals. E_1_ refers to the sum of all transfers from the π MOs of the base to the σ*(M-O) orbitals of the three O atoms adjacent to the base, also associated with the bond paths in Figure 2. The same quantity comprises E_2_ for the O atom that lies opposite to the base. Transfers in the opposite direction, from MO_4_ to base, are contained within the next two columns of Table 2, into the various π* orbitals of the base. E_3_ refers to transfers from MO bonds, while those originating in O lone pairs are contained within E_4_. Some examples of specific NBO orbitals and their mutual intermolecular overlaps are pictured in Figure 3 for the illustrative case of OsO_4_-C_4_H_5_N.

It might be noted first that E_1_ is quite a bit larger than E_2_, if the latter exists at all, consistent with the failure of AIM to locate a bond path to M. The value of E_1_ is roughly consistent with the interaction energy, reinforcing the importance of the bonds to the three adjacent O atoms. There is a modest correlation between these two quantities, with a correlation coefficient of 0.66. The perturbation energies corresponding to the back donation from MO_4_ to the base are generally smaller than E_1_, but certainly far from negligible. With the exception of C_4_H_4_S, E_3_ exceeds E_4_, which indicates the primary source of this back transfer is the set of M-O bonds.

A similar view can be obtained through the use of delocalized canonical MOs. Figure 4 illustrates the relevant frontier MOs of acetylene on the left and RuO_4_ on the right. The red arrow indicates that the energy difference between the HOMO of RuO_4_ and the π* LUMO of acetylene is 0.536 a.u. The size of this gap is diminished to 0.349 a.u., as indicated by the upper blue arrow, if one considers the π HOMO of acetylene and the virtual MO of RuO_4_ whose character most resembles a RuO_opp_ antibonding orbital, resembling that in Figure 3b to some extent. The energy difference is further lowered to only 0.258 a.u. for the lower vacant orbital of the RuO_adj_ sort, suggestive of the localized orbital in Figure 3a. Since the perturbative contribution to the energy is related to the reciprocal of the energy difference, these orbital energy gaps reinforce the notion that the largest component arises from the transfers from the π MO of the acetylene to the RuO_4_, particularly to the antibonding orbital encompassing the adjacent O atoms. The larger gap reduces the contribution in the opposite direction, from the occupied RuO_4_ orbitals to acetylene π*.

The larger share of transfer in the direction of base to MO_4_, than for its reverse, is captured in the positive sign of the overall charge transfer CT. This quantity was computed as the sum of natural charges on the base unit within the complex. CT tends to be largest for the most strongly bound complexes such as those containing C_4_H_5_N, and which typically have the largest values of E_1_. CT is rather small for noncyclic alkyne and alkene, with their weaker bonding and smaller ρ_BCP_.

As a related offshoot of AIM, the reduced density gradient (RDG) can provide an alternate visual interpretation of bonding via a noncovalent interaction (NCI) presentation. The blue and red colors of the surface occurring between the two monomers represent attraction and repulsion, respectively. It is immediately apparent that the blue attractive areas occur between the three O atoms and the atoms of the aromatic ring. In addition to the absence of a bond path to the central M, the corresponding region of the surface are on the cusp between green and yellow, signaling very weak bonding, if any at all. The NCI surfaces of the other complexes closely resemble those in Figure 5.

Another window into the charge shifts that accompany the complexation process can be opened by examining a three-dimensional representation of these shifts. Figure 6 comprises a density difference map, which compares the density of the full OsO_4_-benzene complex with that of the isolated monomers in the precise geometry they occupy within the dimer. The purple areas show where the density is built up by the complexation and depletions are indicated in green, using a threshold of ±0.0002 au for the visualization. The polarizations are obvious in that the benzene shifts electron density up to meet the incoming OsO_4_, so as to be better prepared to shift density from its π-orbitals up to the Os-O antibonds. There is a mirror green polarization within the OsO_4_ that pulls some of its density away from these receptor regions.

The last two columns of Table 2 describe the effects that the complexation with each base has upon the internal bond lengths within MO_4_. These bonds elongate for the most part, but these stretches are rather small. Although the charge transfers into the σ*(M-O_opp_) are fairly small, it is this bond which stretches much more than M-O_adj_. Indeed, there appears to be a fair correlation between the stretch of M-O_opp_ and E_1_, although the latter parameter is related to the O_adj_ atoms.

The nature of these interactions can be further probed through the decomposition of the total interaction energy into several physically meaningful components. The requisite SAPT partitioning yields the values listed in Table 3. In most of these systems, it is the dispersion (DISP) which represents the largest factor. DISP is comparable to the electrostatic component (ES) for the two small bases, but is considerably larger for the aromatic ring systems, sometimes twice as large. Although not negligible, the induction energy (IND) is quite a bit smaller, accounting for only between 6 and 15% of the total of the three attractive elements. Note that IND is largest in magnitude for the C_4_H_5_N complexes which are both most strongly bound, and which also have the largest bond critical point densities.

## 3. Theoretical Methods

The Gaussian 09 program [46] was applied to the systems detailed above. DFT calculations made use of the wB97XD functional, in conjunction with the aug-cc-PVDZ basis set for light atoms. The metal Ru and Os atoms were represented by the triple-valence aug-cc-PVTZ set, with a pseudopotential that accounts for certain relativistic effects. This set was extracted from the Basis Set Exchange (BSE) base group database [47]. It has been demonstrated that wB97XD is more reliable and accurate in studying weak interactions [48,49,50]. Geometries were fully optimized in gas phase with no symmetry restraints and were verified as true minima by the absence of imaginary vibrational frequencies.

The interaction energy E_int_ of each heterodimer was defined as the difference between the energy of the complex and the sum of the energy of the pair of monomers with their geometry frozen in that of the complex. The binding energy E_b_ was obtained by subtracting the energy of the fully optimized monomers from that of the complex. The base set superposition error (BSSE) was corrected using the standard counterpoise protocol [51]. 

Bond paths between the two subsystems, and the densities of the associated bond critical points, were generated via Bader’s Atoms in Molecules (AIM) analysis [52]. Non-covalent interaction (NCI) analysis [53] was used to visualize the weak interactions within the complexes, and the plots were drawn using the Multiwfn [54] and VMD [55] programs. Another means of analysis of the wave function derives from the natural bond orbital (NBO) treatment [56], which invokes a unique set of orthogonal one-electron functions inherent to the *N*-electron wave function first defined by Lowdin. This procedure was used to examine the orbital interactions between electron donor and acceptor. The full interaction energy of each heterodimer was decomposed into physically meaningful components within the framework of symmetry-adapted perturbation theory (SAPT) [57,58], applying the PSI4 program [59].

## 4. Conclusions

The interactions discussed here are reasonably strong, with energies in the range between 9.5 and 26.4 kJ/mol. But there are some questions as to whether these interactions represent a true Osme bond. On one hand, there is a positive σ-hole on the M atom, which can interact with the negative π-region above the plane of the base molecule, and the electrostatic element of each interaction lies in the 13.8–26.1 kJ/mol range. However, there is no identifiable bond path that connects the M with the base unit. The bond paths lead instead from the three O atoms of MO_4_ that lie closest to the base. These three bonds are verified through NCI visualization that also notes only a very weak interaction that includes the M center. The same conclusion arises from NBO, where the bulk of the charge transfer from the π-system of the base to MO_4_ involves the same three adjacent O atoms, rather than the O that lies opposite the base. NBO suggests that the stabilization arising from this transfer is supplemented by back-transfer from the MO bonding orbitals and O lone pairs of MO_4_ to the π* orbitals of the base. Another issue arguing against the presence of a true Osme bond is the large contribution of dispersion to the interaction, coupled with an anomalously small induction component.

## Figures and Tables

**Figure 1 molecules-29-00079-f001:**
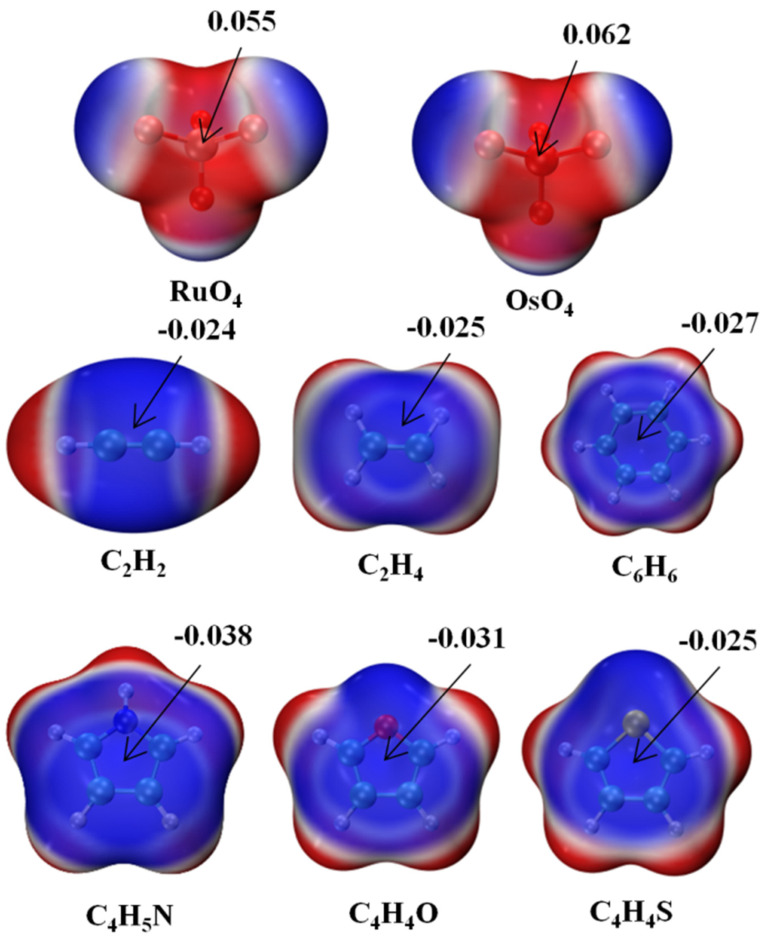
Molecular electrostatic potential (MEP) maps of monomers. Red and blue colors, respectively, indicate the most positive and negative values. Numerical values of extrema in a.u.

**Figure 2 molecules-29-00079-f002:**
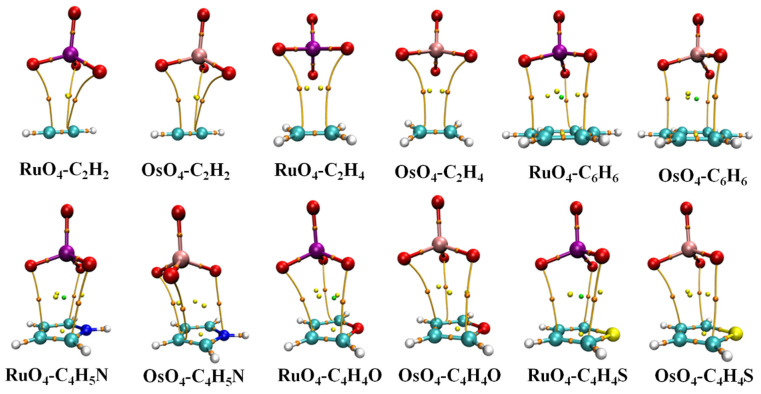
AIM molecular diagrams showing intermolecular bond paths; small red balls indicate position of bond critical point.

**Figure 3 molecules-29-00079-f003:**
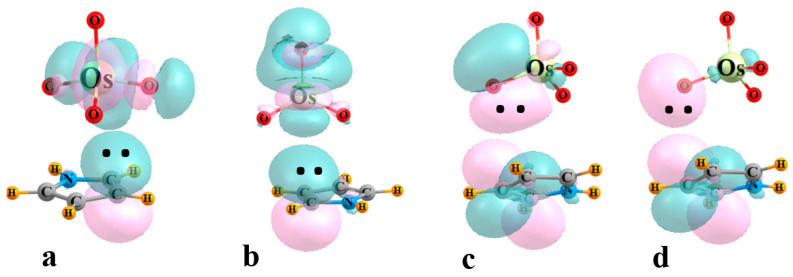
Examples of NBOs involved in charge transfer within the OsO_4_-C_4_H_5_N complex. (**a**) E_1_: π(CC)→σ*(OsO_adj_), (**b**) E_2_: π(CC)→σ*(OsO_opp_), (**c**) E_3_: (OsO)→π*(CC), (**d**) E_4_: O_LP_→π*(CC).

**Figure 4 molecules-29-00079-f004:**
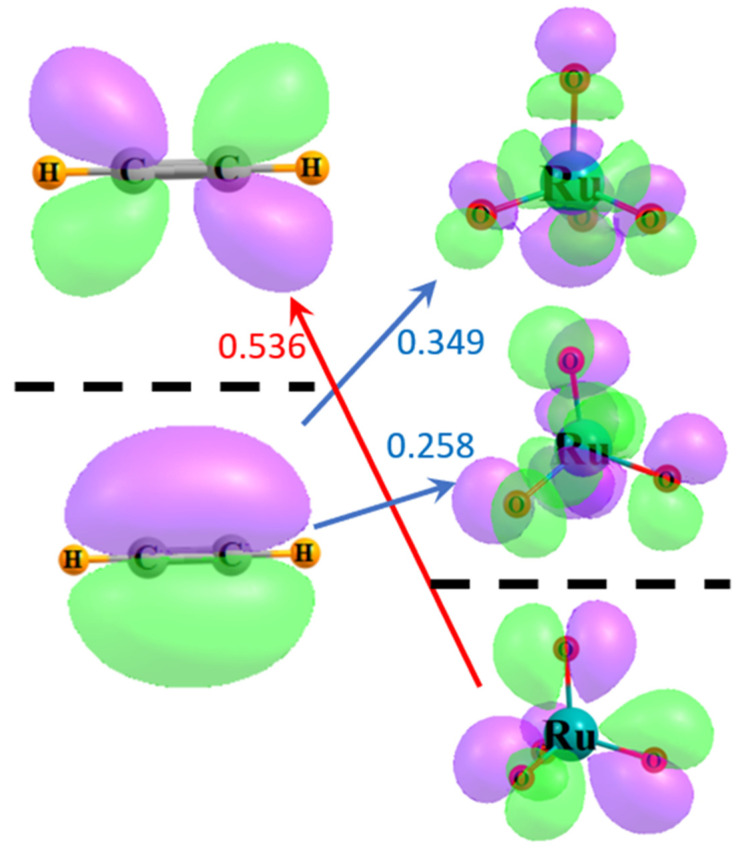
Frontier MOs of acetylene and RuO_4_. Occupied orbitals lie below the broken black line, and virtual orbitals above. Purple and green regions indicate opposite signs within each wavefunction. Numbers refer to difference in orbital energies in a.u.

**Figure 5 molecules-29-00079-f005:**
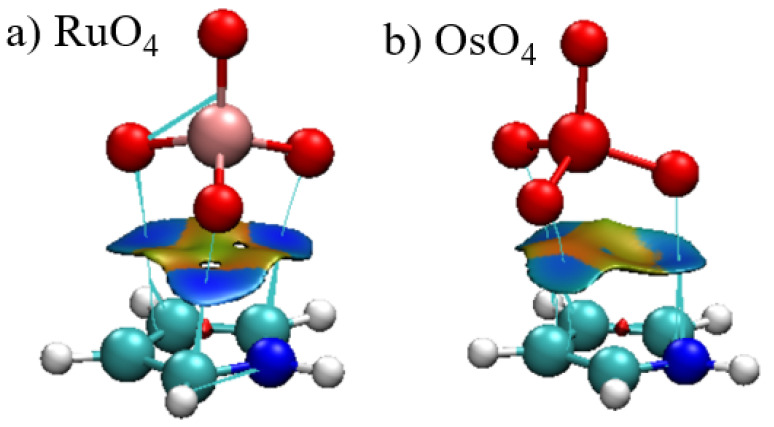
Reduced density gradient visualization of the bonding between C_4_H_5_N and (**a**) RuO_4_ and (**b**) OsO_4_. RDG contour is 0.5 a.u., where blue and red colors represent −0.1 and +0.1 a.u. for ρ·sign(λ_2_).

**Figure 6 molecules-29-00079-f006:**
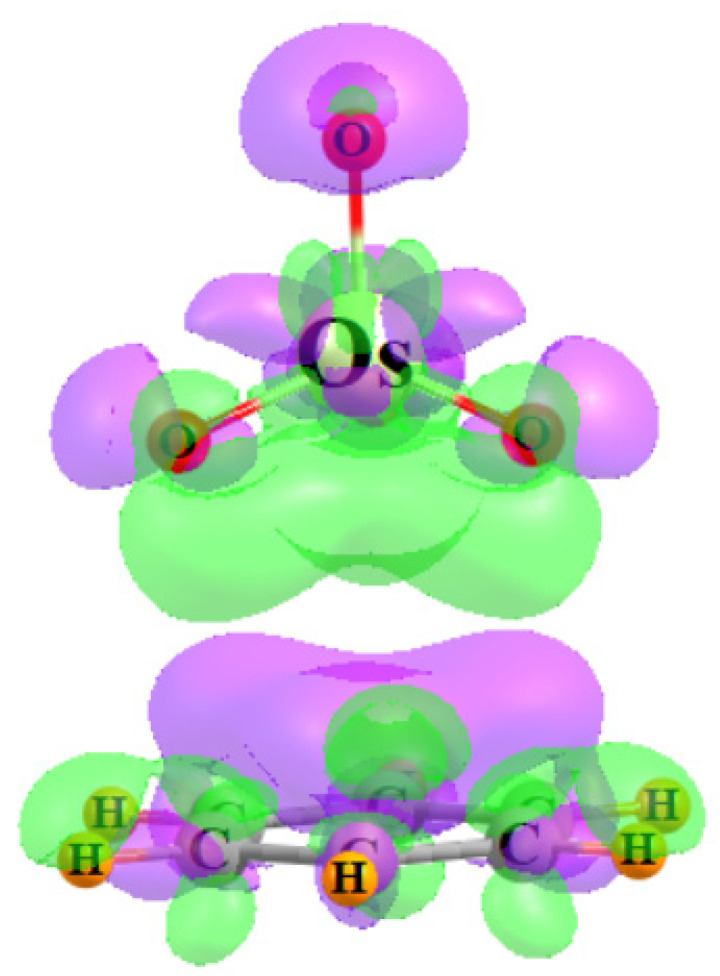
Electron density difference diagram representing the shift of density resulting from the complexation between OsO_4_ and C_6_H_6_. Purple and green colors indicate gain and loss, respectively, by 0.0002 a.u.

**Table 1 molecules-29-00079-t001:** Interaction energy (E_int_, kJ/mol) and binding energy (E_b_, kJ/mol) in complexes, as well as distance R between M atom and center of base unit.

	E_int_	E_b_	R, Å		E_int_	E_b_	R, Å
RuO_4_-C_2_H_2_	−9.5	−9.3	3.684	OsO_4_-C_2_H_2_	−12.7	−12.4	3.671
RuO_4_-C_2_H_4_	−10.5	−10.3	3.709	OsO_4_-C_2_H_4_	−14.6	−14.2	3.660
RuO_4_-C_6_H_6_	−19.3	−18.9	3.667	OsO_4_-C_6_H_6_	−23.2	−23.0	3.763
RuO_4_-C_4_H_5_N	−22.8	−22.6	3.555	OsO_4_-C_4_H_5_N	−26.4	−25.6	3.694
RuO_4_-C_4_H_4_O	−17.1	−17.0	3.658	OsO_4_-C_4_H_4_O	−21.4	−21.1	3.727
RuO_4_-C_4_H_4_S	−18.9	−19.0	3.655	OsO_4_-C_4_H_4_S	−22.1	−21.8	3.775

**Table 2 molecules-29-00079-t002:** Average density of three bond paths (ρ_BCP_, a.u.), second-order perturbation energy (E^(2)^, kJ/mol) of indicated orbital interactions in the complexes, natural charge transfer (CT, e) from base to MO_4_, and internal bond length changes (Å) within MO_4_ caused by complexation.

	ρ_BCP_	E_1_	E_2_	E_3_	E_4_	CT × 10^−2^	Δr(M-O_opp_)	Δr(M-O_adj_)
RuO_4_-C_2_H_2_	0.0053	2.3	0.4	0.4	0.7	0.14	0.0005	−0.0001
RuO_4_-C_2_H_4_	0.0051	2.0	-	0.6	0.2	0.22	0.0004	0.0000
RuO_4_-C_6_H_6_	0.0073	6.5	-	4.3	1.7	2.44	0.0026	0.0002
RuO_4_-C_4_H_5_N	0.0086	11.8	-	5.1	2.7	5.92	0.0041	0.0016
RuO_4_-C_4_H_4_O	0.0070	9.0	0.5	3.5	0.9	2.43	0.0016	0.0004
RuO_4_-C_4_H_4_S	0.0075	6.5	-	2.1	6.0	3.38	0.0026	0.0005
OsO_4_-C_2_H_2_	0.0056	1.9	0.6	1.9	0.5	0.19	0.0001	0.0001
OsO_4_-C_2_H_4_	0.0058	5.7	1.4	2.2	-	0.37	−0.0002	0.0002
OsO_4_-C_6_H_6_	0.0063	6.2	-	1.5	1.2	0.83	0.0015	−0.0002
OsO_4_-C_4_H_5_N	0.0070	10.6	1.9	2.1	0.5	1.82	0.0014	0.0002
OsO_4_-C_4_H_4_O	0.0063	9.8	1.9	2.8	0.2	1.21	0.0008	0.0002
OsO_4_-C_4_H_4_S	0.0063	5.7	2.1	-	1.1	1.30	0.0011	0.0002

E_1_ = Σπ(C-C)→σ*(M-O_adj_); E_2_ = π(C-C)→σ*(M-O_opp_); E_3_ = Σ(M-O)→ π*(C-C); E_4_ = ΣO_LP_→ π*(C-C).

**Table 3 molecules-29-00079-t003:** SAPT components ^1^ of interaction energies (kJ/mol).

	ES	EX	IND	DISP
RuO_4_-C_2_H_2_	−13.8	17.5	−2.2	−16.2
RuO_4_-C_2_H_4_	−14.2	18.9	−2.1	−17.5
RuO_4_-C_6_H_6_	−19.8	37.9	−8.4	−41.7
RuO_4_-C_4_H_5_N	−25.7	45.4	−12.5	−42.5
RuO_4_-C_4_H_4_O	−19.2	32.9	−6.7	−34.9
RuO_4_-C_4_H_4_S	−20.1	37.4	−8.5	−41.4
OsO_4_-C_2_H_2_	−16.0	19.3	−2.8	−16.0
OsO_4_-C_2_H_4_	−18.0	23.1	−3.2	−18.2
OsO_4_-C_6_H_6_	−17.1	29.5	−4.9	−34.3
OsO_4_-C_4_H_5_N	−26.1	36.0	−7.4	−34.6
OsO_4_-C_4_H_4_O	−19.6	30.1	−5.4	−30.9
OsO_4_-C_4_H_4_S	−18.3	30.3	−5.5	−33.9

^1^ Computed with def2-TZVP basis set.

## Data Availability

Data is available from the authors on request.

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
