# Peer review of "Search for Osme Bonds with π Systems as Electron Donors"

_molecules, 2023, doi:10.3390/molecules29010079_

Round 1
Reviewer 1 Report
Comments and Suggestions for Authors
In my opinion, this is a very interesting but also quite controversial work.
I have the impression that OsO4-π system interactions should be mainly viewed as σ (Ru-O) - π*(C-C) interaction. This is primarily evidenced by the data of QTAIM analysis. On the other hand, NBO results indicated that the most significant contribution is made by π(C-C)-σ *(Ru-O) interactions. Perhaps, analysis of canonical molecular orbitals can provide additional information to this conundrum.
Did the authors try to orient heterocyclic molecule (C5HN, C4H4O, C4H4S) in such a way that a potential M(Os,Ru)-X(N,O,S) bond is formed? I suppose, that in this case osme bonds must be observed.
Why did authors use only electron density at bcp as the one and only descriptor of QTAIM analysis?
Is there a possibility of incorrect characterization of the electron density distribution when using effective core potentials? Can this affect on QTAIM analysis data?
After answering these questions, which are mostly debatable, the work may be accepted for publication.
Comments on the Quality of English Language
I suppose that generally quality of the manuscript writing is fine.
Author Response
I have the impression that OsO4-π system interactions should be mainly viewed as σ (Ru-O) - π*(C-C) interaction. This is primarily evidenced by the data of QTAIM analysis. On the other hand, NBO results indicated that the most significant contribution is made by π(C-C)-σ *(Ru-O) interactions. Perhaps, analysis of canonical molecular orbitals can provide additional information to this conundrum.
Reply: We considered delocalized canonical MOs in paragraph 7 of the result section to further explain this problem.
Did the authors try to orient heterocyclic molecule (C4H5N, C4H4O, C4H4S) in such a way that a potential M(Os, Ru)-X(N,O,S) bond is formed? I suppose, that in this case osme bonds must be observed.
Reply: In the previous studies (Angew. Chem. Int. Ed. 2021, 133, 20891; Inorganics 2022, 10(9), 133), such configuration with N/O/S---M interaction has been confirmed with similar molecules including lone pairs on these atoms. However, we focus only the π---M interaction in this paper.
Why did authors use only electron density at bcp as the one and only descriptor of QTAIM analysis?
Reply: Yes, we only used electron density at bcp to estimate the strength of Osme bond. This value is an average of electron densities of two or three BCPs, which is always positive so an average is meaningful. However, both Laplacian and energy density may be positive or negative, making their average of questionable value.
Is there a possibility of incorrect characterization of the electron density distribution when using effective core potentials? Can this affect on QTAIM analysis data?
Reply: There are many studies to perform the AIM analyses for the non-covalent interactions using effective core potentials. We tested AIM analysis with an all-electron basis set for Ru or Os, and found very small perturbations.
Reviewer 2 Report
Comments and Suggestions for Authors
This manuscript deals with a detailed investigation of interaction of RuO4 and OsO4 with selected electron donors. The calculations were performed at appropriate level of theory, the manuscript is nicely written and I suggest its acceptance.
Author Response
Reply: Thanks for this positive comment and suggestion for acceptance.
Reviewer 3 Report
Comments and Suggestions for Authors
Scheiner and coworkers described theoretical considerations about the nature of some osmium chemical bonds. This topic is interesting, and the manuscript is written well. Some improvements are hovewer necessary. In parcicular:
- The application of the wB97XD functional should be justified using respective examples from the recent literature.
- The quality of maps on Figure 1 is not sufficient.
- Views on figure 2 are undreadable.
- Table 1,2,3 and the discussion on the text: energies should be specified with the one decimal place.
- Charge transfer (CT) should be specified with two decimal places. The methodology of the estimation of CT should be specified.
- Do solvatation model was implemented in the calculations?
Author Response
- The application of the wB97XD functional should be justified using respective examples from the recent literature.
Reply: In the fourth paragraph of the introduction and in the methods section, we have cited some relevant papers in the literature using wB97XD functional.
- The quality of maps on Figure 1 is not sufficient.
Reply: The picture has been modified appropriately. The reviewer is not clear as to what aspect of this figure is insufficient. We and many other authors have used figures of this type to indicate positive and negative regions of the potential. Without more guidance we are at a loss as to what it is the reviewer requests us to do.
- Views on figure 2 are unreadable.
Reply: Again, the reviewer is not clear as to what makes this figure unreadable in their opinion. The atoms are clearly indicated, as are the bond paths, and the locations of the critical points are shown as the small red balls, as explained in the figure. Again, we would be happy to alter this diagram, but require guidance as to what is requested.
- Table 1,2,3 and the discussion on the text: energies should be specified with the one decimal place.
Reply: The relevant data has been re-recorded with the one decimal place.
- Charge transfer (CT) should be specified with two decimal places. The methodology of the estimation of CT should be specified.
Reply: The number of CT decimal places has been reduced to two. The methodology used to estimate this quantity has been defined in the text where it is described “as the sum of natural charges on the base unit within the complex”.
- Do solvatation model was implemented in the calculations?
Reply: Correlation calculations are performed in a gas phase environment. This has been mentioned in the Methods section that calculations were gas phase with no inclusion of solvation effects.
Round 2
Reviewer 1 Report
Comments and Suggestions for Authors
All necessary corrections have been done, thus manuscript is suitable for publication.